# Mode of Action of a Novel Synthetic Auxin Herbicide Halauxifen-Methyl

**Jiaqi Xu [1,2], Xudong Liu [1,2], Richard Napier [3] , Liyao Dong [1,2] and Jun Li [1,2,*]**

1    College of Plant Protection, Nanjing Agricultural University, Nanjing 210095, China;
     2018102123@njau.edu.cn (J.X.); 2020102121@stu.njau.edu.cn (X.L.); dly@njau.edu.cn (L.D.)
2    Key Laboratory of Integrated Management of Crop Diseases and Pests, Ministry of Education,
     Nanjing Agricultural University, Nanjing 210095, China
3    School of Life Sciences, University of Warwick, Gibbet Hill Road, Coventry CV4 7AL, UK;
     richard.napier@warwick.ac.uk
*    Correspondence: li_jun@njau.edu.cn

**Abstract:** Halauxifen-methyl is a new auxin herbicide developed by Corteva Agriscience (Wilmington, DE, USA). It has been suggested that *ABF5* may be the target of halauxifen-methyl, as *AFB5* mutants of *Arabidopsis thaliana* are resistant to halauxifen-methyl, which preferentially binds to *AFB5*. However, the mode of action of halauxifen-methyl has not yet been reported. Therefore, the aim of the present study was to reveal the mode of action of halauxifen-methyl by exploring its influence on indole-3-acetic acid (IAA) homeostasis and the biosynthesis of ethylene and Abscisic Acid (ABA) in *Galium aparine*. The results showed that halauxifen-methyl could disrupt the homeostasis of IAA and stimulate the overproduction of ethylene and ABA by inducing the overexpression of the 1-aminocyclopropane-1-carboxylate synthase (*ACS*) and 9-cis-epoxycarotenoid dioxygenase (*NCED*) genes involved in ethylene and ABA biosynthesis, finally leading to senescence and plant death.

**Keywords:** halauxifen-methyl; mode of action; auxin homeostasis; ethylene; abscisic acid

## 1. Introduction

Synthetic auxin herbicides (SAHs) include a remarkable suite of chemical compounds that preferentially have profound morphological effects on growing dicot weeds, ultimately leading to plant death [1]. SAHs have a unique mode of action that mimics the function of indole acetic acid (IAA) [2]. They exhibit systemic mobility and selectivity, and already to be one kind of predominant herbicides to control weeds in cereal crops. 2,4-dichlorophenoxyacetic acid (2,4-D) is among the most auxin-active molecules and was commoditized as the first auxin herbicide in the 1940s.The commercialization and popularization of 2,4-D marked the beginning of a new era of weed control in modern agriculture. Since then, various chemical classes of auxin herbicides, with different herbicidal spectra and modes of action, have been synthesized, commoditized, and become effective weed control tools [2]. At present, there are approximately 20 commercial compounds and even more experimental compounds that are classified as auxin herbicides, all of which contain an aryl group with an attached carboxylic acid functionality, and they can be divided into several categories based on their chemical structure differences [1]. Currently, synthetic auxin herbicides are classified into phenoxycarboxylic (2,4-D), benzoic (dicamba), pyridinecarboxylic (fluroxypyr), and quinolinecarboxylic (quinclorac) acids [2]. In the past few years, new auxin herbicides, such as pyrimidine carboxylic acids, aminocyclopyrachlor, and 6-aryl-picolinate herbicides, such as halauxifen-methyl (ArylexTM active), have been developed [1]. It is remarkable that new synthetic auxin herbicides are still being introduced today, which indicates that this unique plant-specific mode of action (MoA) is still valuable and relevant to modern agriculture.

Indole-3-acetic acid (IAA), the endogenous auxin, plays an important role in all aspects of plant development. It can promote growth and germination at low concentrations, while the opposite is true when it is used at high concentrations [3]. The effects of synthetic auxin herbicides on plants are similar to those induced by excessive treatment with the natural plant hormone auxins, such as IAA [2]. The perception and signaling pathway of endogenous auxin was identified a long time ago [4–6]. In contrast, although synthetic auxin herbicides have been used for more than 70 years, their precise mode of action is not fully known. The perception and signaling pathway of endogenous auxin is considered to trigger plant death through auxin herbicides [2,7]. Through the investigation of *Arabidopsis thaliana* auxin-resistant mutant lines, the auxin receptor and signaling pathways that are essential for the plant perception and specificity of auxin herbicides were discovered, revealing the ligand/receptor system [8]. The *TIR1/AFB* gene family consists of six receptors: *TIR1* and five homologs of *AFB* [9–11]. Auxin perception and signaling is conceptually straightforward: substrate receptor binding leads to the degradation of inhibitors and the activation of transcription factors. Analysis of *Arabidopsis thaliana* mutant lines has proved that auxin receptor genes are involved in plant perception and contribute to the specificity of auxin herbicides. For example, the *TIR1* mutants are resistant to dicamba and 2,4-D [12]. In addition, the in vitro auxin receptor binding studies show that auxin herbicides can bind with auxin receptors, such as picloram interacting preferentially with *AFB5* [13]. The *TIR1/AFB* receptors' link-binding of auxin herbicides directly leads to the activation of transcription factors and overexpression of auxin-responsive genes, which in turn cause a subsequent series of biochemical and physiological events related to the action of auxin herbicides [2]. Amongst the auxin-response genes is the family of ACC synthases (*ACS*). Auxin herbicides induce overexpression of *ACS* genes, resulting in an increase in ACC synthase activity and eventually leading to an increase in ethylene formation. Kraft et al. [14] observed increased expression of *ACS* and an increase in ethylene levels in *Galium aparine* after treatment with auxin herbicides. In addition, ABA accumulation was found in a variety of dicot species after treatment with auxin herbicide [15]. The key step in ABA biosynthesis is the oxidative cleavage of 9-cis-epoxide carotenoid to xanthine aldehyde, which is catalyzed by 9-cis-epoxycarotenoid dioxygenase (*NCED*); the enzyme is encoded by a family of *NCED* genes [16,17]. Kraft et al. [14] found that auxin herbicides can upregulate gene expression of *NCED* and abscisic acid accumulation in the shoot tissue of *Galium aparine*. Excessive ethylene and ABA are thought to be the primary mode of action of auxin herbicides [14,15,18,19].

Halauxifen-methyl is one of the 6-aryl picolinates, which are a new family of auxin herbicides [20] developed by Corteva Agriscience. The structure of these novel auxin herbicides, which contain a new aryl picolinate structure is built on the picolinic acid scaffold and the 6-aryl group makes it obviously different from the structure of all the other auxin-type herbicides listed above [21]. Due to its low soil mobility, low use rate and short soil half-life, halauxifen-methyl can be widely used with less use restrictions than other auxin-type herbicides [22,23]. Halauxifen-methyl can be absorbed by the leaves, translocate systemically through the phloem and xylem stream, and finally accumulate in the meristematic tissue [22]. When used as a herbicide, the symptoms caused by halauxifen-methyl in susceptible plants are similar to those caused by other auxin herbicides, including epinasty, deformation, necrosis, and eventual plant death [22]. According to the study by Dow AgroSciences, *AFB5* mutants of *Arabidopsis thaliana* are resistant to halauxifen-methyl, suggesting that *AFB5* may be the target of halauxifen-methyl [1]. Another helped confirm that the finding that halauxifen-methyl preferentially bound to *AFB5* over *TIR1* in SPR binding studies [1]. However, there are few studies on the mode of action of this novel herbicide post receptor. In *Brassica napus*, halauxifen-methyl treatment leads to the upregulation of auxin and hormone responses, such as IAA, ABA, and ACC concentration [23]. McCauley et al. [24] found that halauxifen-methyl enhances the expression of NCED and leads to a rapid biosynthesis of ABA in *Erigeron canadensis*. In the present study, the influence of halauxifen-methyl on the IAA homeostasis and ethylene and ABA biosynthesis of *Galium*

*aparine*, which is a common dicotyledonous weed in wheat fields that is susceptible to halauxifen-methyl on this common dicotyledonous weed of wheat fields [25].

## 2. Materials and Methods

### 2.1. Plant Materials and Cultivation of Plants

*Galium aparine* seeds for experiments were collected in the summer of 2017 from wheat fields in Minhe Village, Jiangdu Fairy Town, Yangzhou City, Jiangsu Province, China. Dormancy-broken seeds were pregerminated in open trays in the illumination incubator (25 °C). They were then germinated in vermiculite substrate moistened with clear water in an illumination incubator (day/night: 14/10 h at 25/20 °C). When seedlings grew to the first whorl stage, they were transferred to half-strength Linsmaier–Skoog nutrient solution and grown to the three-whorl stage in an illumination incubator (day/night: 14/10 h at 25/20 °C). When the plants grew to the three-whorl stage, the uniformly developed plants were transferred into 320 mL plastic cups with half-strength Linsmaier–Skoog medium (each cup contained 10 plants), and then the plastic cups were placed in an illumination incubator (day/night: 14/10 h at 25/20 °C). The solution was changed every three days. After a week of adaptation, halauxifen-methyl (final concentrations were 0.5, 5, and 50 μM, respectively) and IAA (final concentrations were 1, 0.1, and 0.01 mM) were added to the medium in N,N-dimethylformamide (DMF) (0.1% final concentration of DMF). At various times (0, 6, 12, 24 h) after treatment, the shoots from parallel cups were harvested, immediately frozen in solid liquid nitrogen, and then stored at −80 °C.

### 2.2. Determination of Ethylene Production

The influence of halauxifen-methyl on ethylene production of *Galium aparine* was examined using the following method [26]. After treatment with halauxifen-methyl in hydroponic solution, the fresh weight (FW) of treated plants was measured, and then they were transferred into 20 mL head space bottles with 200 μL ultrapure water in them (one plant per bottles; three replications). The bottles with plants were sealed with metal caps that were covered with septa. After incubation for a further 3 h in the illumination incubator (25 °C), a 1 mL gas sample of the head space was taken from each bottle and measured immediately with a gas chromatograph (GC9790Plus, Fuli Analytical Instrument Co., Ltd., Zhejiang, China) equipped with a flame ionization detector and a 30 m × 0.32 mm × 0.25 mm $Al_2O_3$ column. The column temperature was 50 °C, the injector temperature was 150 °C, the carrier gas flow was set to 90 mL $min^{-1}$, and the oxidant gas flow was set to 75 mL $min^{-1}$.

### 2.3. Determination of ACC Content

A previous methodology [27] with slight modifications was used to measure the ACC content in *Galium aparine*. A total of 200 mg of plant material was powdered in liquid nitrogen and then extracted with 70% (*v/v*) aqueous ethanol. In order to remove the ethanol, the extract was centrifuged for 10 min at 10,000 rpm at 4 °C. The supernatant was passed through a 0.2 μM filter. Following this, the supernatant was converted to ethylene and then quantified using gas chromatography.

### 2.4. Determination of ACC Synthase Activity

The ACC synthase activity of the treated *Galium aparine* was measured using the following method [25]. After being powdered under liquid nitrogen, 200 mg of plant material was homogenized in 2 mL 100 mmol·$L^{-1}$ potassium phosphate buffer (pH 8.5), which contain dithiothreitol (5 mM), leupeptin (10 μM), and pyridoxal phosphate (6 μM). The extract was centrifuged for 10 min at 10,000 rpm at 4 °C and the supernatant was passed through a Sephadex G25 column. Subsequently, 0.3 mL crude extraction liquid was mixed with a 0.3 mL *ACS* assay mixture (in 80 mM potassium phosphate buffer (pH 8.5)) containing 20 μM PLP and 100 μM SAM. After two hours of incubation at 37 °C, 20 μmol of mercury (II) chloride was added to stop the reaction. Then, the ACC produced

was quantified through chemical conversion to ethylene. The ACC synthase activity was described as the ACC production rate. The background level of the ACC can be measured by converting it to ethylene prior to the reaction.

### 2.5. Determination of ACC Oxidase Activity

ACC oxidase was extracted and assayed as described by Dupille and Zacarías [28]. After weighing, individual treated plants were carefully transferred into 5 mL glass vials with plastic caps which contained 3 mL of 5 mM ACC solution (in 25 mM potassium phosphate buffer (pH 5.3)), and the vials were sealed with septa. After incubation at 25 °C in darkness for 1 h, a 1 mL gas sample of the head space was obtained for ethylene measurements using gas chromatography. The ACC oxidase activity is expressed in terms of the ethylene production rate.

### 2.6. Determination of IAA and ABA

For IAA and ABA determination, 5 g of plant material was powdered in liquid nitrogen and then extracted with 80% (*v/v*) aqueous methanol containing 1 mM butylated hydroxytoluene (BHT) over 12 h (three replicate extractions). To remove the ethanol, the extract was centrifuged for 15 min at 10,000 rpm at 4 °C. The residue was extracted with 80% (*v/v*) aqueous methanol again. The supernatant was combined and passed through a 0.2 μM filter. The volume fraction of methanol was adjusted to 33% by diluting the sample extract with distilled water, after which ammonia was used to adjust the pH of the sample extract to 8.5. Following this, the sample extract was passed through an SPE column (MAX, Thermo Fisher Scientific, Shanghai, China), which had been equilibrated with 5 mL of methanol and 5 mL of 2% ammonia spirit. The column was washed with 5 mL of 2% ammonia spirit and 5 mL of methanol. Then, 5 mL of 1% formic acid-methanol was applied as an elution solvent to the MAX column and the efflux was collected, which contained phytohormones with neutral and acidic characters: IAA and ABA. The efflux was concentrated to dryness in a Termovap sample concentrator and dissolved in 300 μL methanol for high performance liquid chromatograph (HPLC) analysis. For the HPLC conditions, the chromatographic column was a $C_{18}$ column (Agilent, $4.6 \times 250$ mm) and the column temperature was room temperature. The run gradients were: A—0.6% formic acid in ultrapure water, B—100% methanol, and C—100% acetonitrile; A:B:C = 55:40:5 (*v/v/v*). The flow rate was 1 mL/min, the UV detection wavelength was set at 269 nm, and the injection volume was 20 μL.

### 2.7. Molecular Cloning of the GaACS4, GaACS7, and GaNCED1 Fragments

The *GaACS4* gene fragment was cloned based on homology to GenBank entries for *ACS4* from *Arabidopsis thaliana* (accession NM_127846), *Solanum tuberosum* (accession XM_006345517), *Solanum lycopersicum* (accession NM_001246946), *Pisum sativum* (accession KX255646), *Momordica charantia* (accession FJ459814), and *Ricinus communis* (accession DQ300359). Based on the sequences conserved between the various cDNA clones, a pair of primers (forward ATGGGTCTTGCGGA-AAATCA and reverse GCGAAACAAACTC-TAAACCA) were designed. The primers of *GaACS7* (forward CAGATGGGATTGGCA-GAAAAT, reverse CAAAGCAAACC-CTGAACCAACC) and *GaNCED1* (forward CG-CAATTACTGAGAACTTCGTC, reverse CGAGTTTGTTTCGGTTCACCATTC) were designed as described above [14]. The PCR conditions were 95 °C for 3 min and 35 cycles of 95 °C for 15 s, 52 °C for 15 s, and 72 °C for 1 min, then 72 °C for 10 min. After DNA sequencing, the resulting fragment was cloned into NCBI BLAST and confirmed for homology with the *ACS4*, *ACS7*, and *NCED* genes from other plants.

### 2.8. Gene Expression Analysis Based on Real-Time Quantitative PCR

Based on the DNA fragments obtained above, the primers of *GaACS4* (forward T-CCAGAAATACAGCCCTGCA, reverse GACCCAAACACAGCGCTTAA), GaACS7 (forward CTTGACCAACCCTTCGAACC, reverse TTGCACTCAACGTCGTCTTC), and *GaNCED1*

(forward TGATTTCCCCGTCCTTGTGT, TGGCGAGGTTTGGAGT-TTTG), for the real-time quantitative PCR (qRT-PCR) were respectively designed using Primer Premier 5.0 software. Total RNA was isolated from the shoot tissue of treated *Galium aparine* using an RNAsimple Total RNA Kit (TIANGEN, Beijing, China) and reverse transcribed into cDNA using a PrimerScriptTM RT Reagent Kit (Vazyme, Nanjing, China). The Ga28S gene, which is stably expressed in many plant tissues and under various stress conditions was selected as the reference sequence [29]. The primers of *Ga28S* (forward TTGTCCGCAT-CAAAACTGGG, AACGACTAT-TCCGGCACTCT) for the qRT-PCR were obtained from the report by Su et al. [29]. QRT-PCR reactions were performed with 20 µL volumes using ChamQ SYBR qPCR Master Mix (Vazyme, Nanjing, China), each containing 10 µL of SYBR Green Supermix, 0.4 µL of 10 µM primers (F/R), 0.4 µL of Rox, 2 µL of diluted cDNA, and 6.8 µL of nuclease-free $H_2O$. The qRT-PCR reaction conditions included one cycle of 30 s at 95 °C, 40 cycles of 10 s at 95 °C, and 30 s at 60 °C. Relative transcript levels were calculated using the $2^{-\Delta\Delta Ct}$ method.

## 3. Results

### 3.1. Effect of Halauxifen-Methyl on Growth of Galium aparine

Within 24 h of treatment with halauxifen-methyl, various symptoms could be observed, such as leaf epinasty, tissue swelling, and stem twisting, similar to those that occurred after application with excessive IAA. Five days after application, the growth inhibition was more obvious than that after 24 h, and even senescence or death could be observed. At this point, we measured the fresh weight of the whole plants after different treatments. The results showed that the fresh weight of the *Galium aparine* plants was significantly reduced after the use of halauxifen-methyl, and the degree of reduction was related to the concentration of the herbicide (Figure 1).

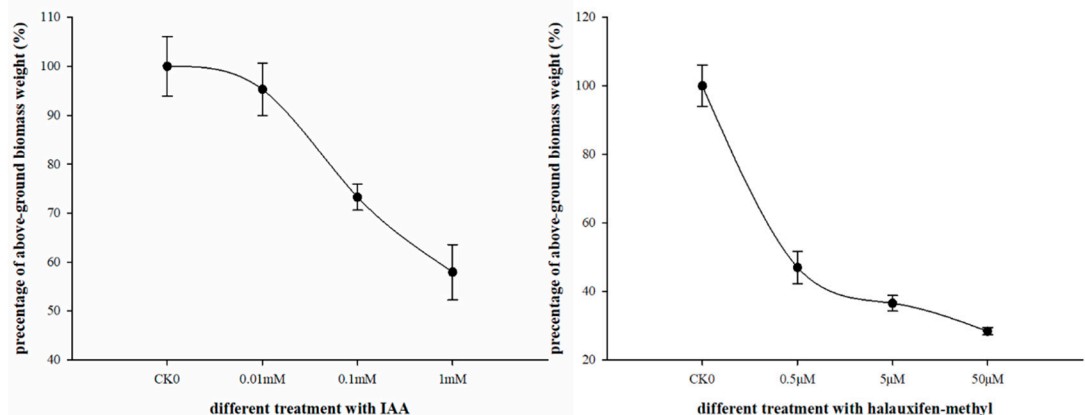

**Figure 1.** The effects of halauxifen-methyl and IAA at different concentrations on the above-ground biomass of *Galium aparine*. Data are expressed as percentages of the mean values for untreated control plants. Vertical bars represent the standard errors of the means.

### 3.2. Effect of Halauxifen-Methyl on the Content of IAA in Galium aparine

*G. aparine* were treated with 0.5, 5, and 50 µM of halauxifen-methyl, respectively, by applying it to the nutrient solution. IAA levels in the shoots continued to increase within 24 h after treatment and were 1.9-fold, 2.4-fold, and 2.9-fold higher than those in the control, respectively (Figure 2).

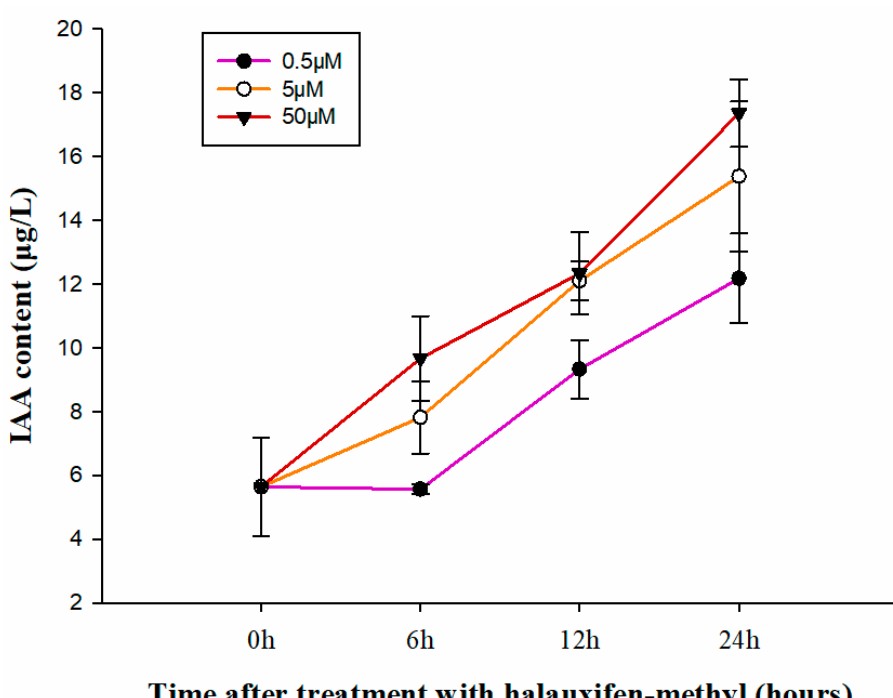

**Figure 2.** The effects of halauxifen-methyl on IAA levels in shoots of *G. aparine*. Data are expressed as percentages of the mean values for untreated control plants. Vertical bars represent the standard errors of the means.

### 3.3. Effect of Halauxifen-Methyl on Ethylene Biosynthesis in Galium aparine

To determine the relationship between the herbicidal action of halauxifen-methyl and ethylene biosynthesis in *Galium aparine*, ethylene production capacity, ACC contents, ACC synthase activity, and ACC oxidase activity in *Galium aparine* were measured after halauxifen-methyl treatment. After treatment with halauxifen-methyl, ethylene production in *Galium aparine* continued to increase over the first 12 h and slightly decreased later. In the first 24 h, the maximum ethylene production increased to respectively 5-fold, 6.8-fold, and 11-fold higher than the levels in control plants (Figure 3A). Treatment with halauxifen-methyl resulted in a concentration-dependent increase in ACC levels in *Galium aparine* within 12 h (Figure 3B). The time course showed that maximum ACC levels were reached at 12 h and they then decreased slightly. At 12 h after treatment, the ACC levels were 3.7, 6.3, and 8.1 times higher than the control. The time course for the ACC synthase activity in the treated plants was similar to that for the ACC levels. ACC synthase activity reached its peak at 12 h after treatment, at which point the enzyme activity increased to levels 3.3, 4.8, and 8.5 times as high as that of the control (Figure 3C). As for the ACC oxidase activity, it continued to increase after treatment with 0.5 and 5μM halauxifen-methyl for the first 24 h, while it increased at first and then decreased after treatment with 50 μM halauxifen-methyl. The maximum ACC oxidase activity levels were respectively 3.4-fold, 4.8-fold, and 6.1-fold higher than that of the control (Figure 3D). The results reveal that halauxifen-methyl treatment can stimulate the activities of ACC synthase and ACC oxidase, key rate-limiting enzymes in ethylene synthesis, in a short time, thus leading to an increase in ethylene precursor ACC and ethylene production.

To further determine the effects of halauxifen-methyl on the ethylene biosynthesis pathway of *Galium aparine*, the gene expression of the *GaACSs* that encode the key rate-limiting enzymes for ethylene biosynthesis were measured after treatment with 5 μM of halauxifen-methyl. Studies have shown that exogenous application of IAA, ethylene, and ACC can increase the expression of *ACS4* and *ACS7* in Arabidopsis [30,31], based on which we hypothesized that the synthetic auxin herbicide halauxifen-methyl, which has a similar effect to IAA, can increase the expression of *ACS4* and *ACS7* genes as well. Therefore, we

selected *GaACS4* and *GaACS7* as experimental genes. The results showed that expression levels of *GaACS4* and *GaACS7* continued to increase within 24 h of halauxifen-methyl treatment up to a maximum of respectively 6.8-fold and 8.1-fold greater than in controls at 24 h (Figure 4).

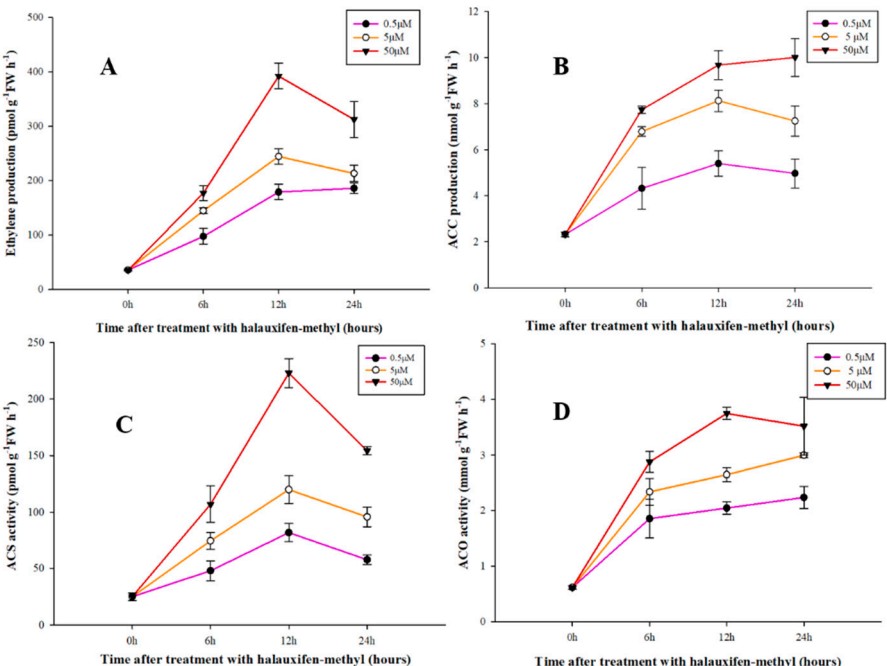

**Figure 3.** The effect of halauxifen-methyl on the ethylene production capacity (**A**), ACC contents (**B**), ACC synthase activity (**C**), and ACC oxidase activity (**D**) of the treated *Galium aparine*. Data are expressed as percentages of the mean values for untreated control plants. Vertical bars represent the standard errors of the means.

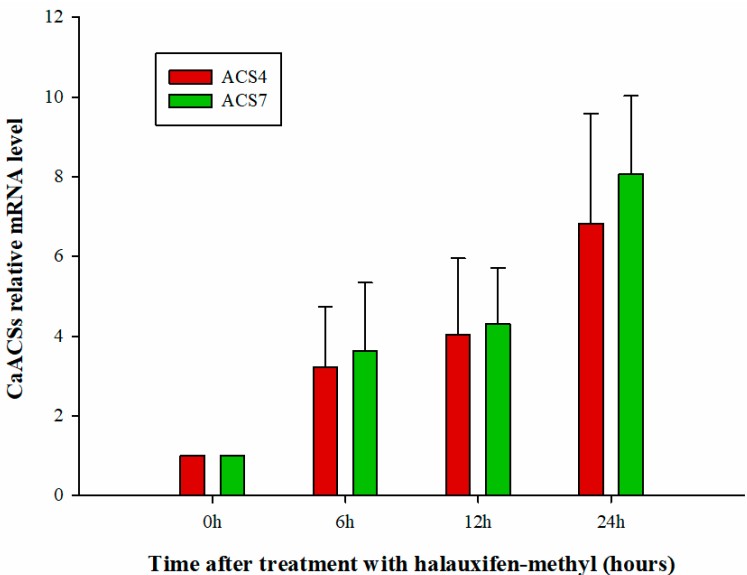

**Figure 4.** The *GaACS4* and *GaACS7* gene expression patterns in *G. aparine* at 0, 6, 12, and 24 h after treatment with 5 μM of halauxifen-methyl. Vertical bars represent the standard errors of the means.

*3.4. Effect of Halauxifen-Methyl on ABA Biosynthesis in Galium aparine*

Abscisic acid is another of the five plant hormones involved in plant responses to abiotic stress, stomatal closure, and regulation of senescence. Several authors have observed that the application of auxin herbicides can lead to the accumulation of ABA in

plants [14,24,32]. In this experiment, we measured the changes in ABA content in *Galium aparine* at 6, 12, and 24 h after treatment with halauxifen-methyl. The results showed that the ABA contents continued to increase within the first 24 h after treatment with halauxifen-methyl (Figure 5).

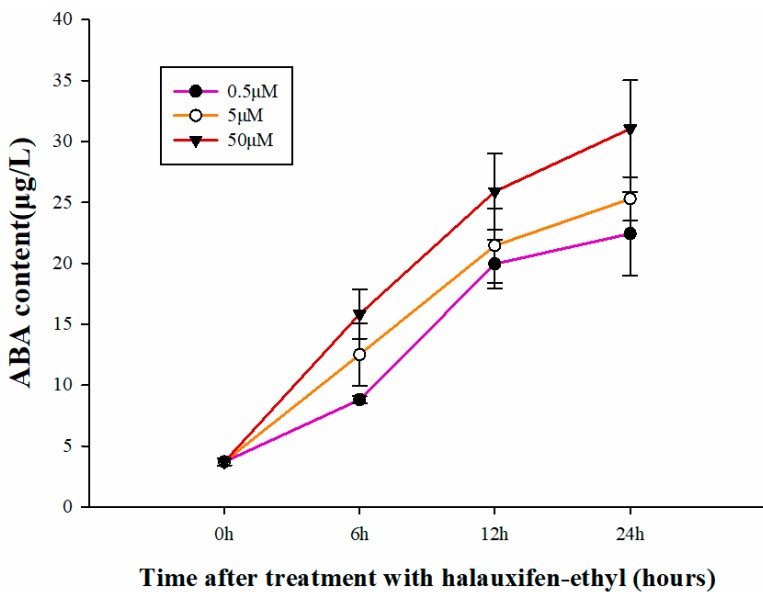

**Figure 5.** The effects of halauxifen-methyl on ABA levels in shoot tissue of treated *Galium aparine*. Vertical bars represent the standard errors of the means.

ABA increases induced by auxin have generally been found to be due to xanthophyll cleavage [33]. 9-cis-epoxycarotenoid dioxygenase (*NCED*) is the key rate-limiting enzyme in ABA biosynthesis and is involved in xanthophyll cleavage [34–36]. *NCED* is encoded by the *NCED* gene family [37,38], and after treatment with 5 µM of halauxifen-methyl, the *GaNCED1* gene in *Galium aparine* was consistently upregulated in the first 24 h (Figure 6). The herbicide treatment at 24 h resulted in the highest gene expression level for *GaNCED1*, which was 14.4-fold higher than in controls.

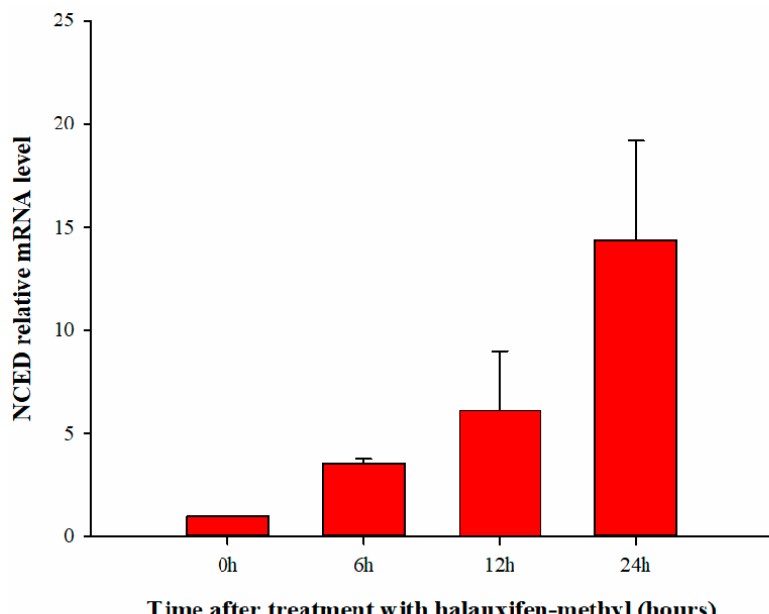

**Figure 6.** The *GaNCED1* gene expression patterns in *Galium aparine* at 0, 6, 12, and 24 h after treatment with 5 µM of halauxifen-methyl. Vertical bars represent the standard errors of the means.

## 4. Discussion

Synthetic auxin herbicides (SAHs) are one of the most widely used herbicides worldwide [2]. Halauxifen-methyl is a new SAH representing a new chemical class [23], and it has been widely used in China for weed control in paddy rice fields, especially for *Echinochloa species*.

In general, the mode of action of SAHs involves over-induction of auxin response in susceptible plants, such as production of ethylene and ABA. It is well-reported that upregulation of ethylene biosynthesis accounts for a large part of the repertoire of SAH-mediated responses, and it is a typical early reaction in various weeds after treatment with SAHs, such as in *Echinochloa crusgalli* var. *zelayensis* after treatment with quinclorac or *Galium aparine* with dicamba and picloram, respectively [2,4,39–41]. Following overproduction of ethylene, ABA accumulates in plants [26,39,41]. Several studies have shown that ABA plays an important role in the modes of action of various auxin herbicides [14,24,26,42]. In the present study, overproduction of ethylene and accumulation of ABA were also observed in *G. aparine* when treated with halauxifen-methyl, which suggested that halauxifen-methyl may share the same set of phytotoxic responses and extending range of weed families shown to respond to these new herbicides.

Application of SAHs lead to auxin overdose which, like excess endogenous auxin concentrations is likely to lead to the imbalance of auxin homeostasis [2]. Studies have shown that 2,4-D treatment, for example, may cause either a decrease or an increase level of free IAA in plants [43]. In the case of G. aparine treated with halauxifen-methyl, it was shown that endogenous IAA levels rise within 6 h and continue to rise for 24 h (Figure 2). At the lowest dose of halauxifen-methyl, the rise in endogenous IAA appears slower, but in all cases at all doses the plants are overwhelmed by the additional load of the synthetic auxin in the treatment. The combined rise in auxins induced dramatic rises in ethylene and ABA concentrations in G. aparine (Figures 3A and 5). The ethylene concentration rose more than three-fold within 6 h, and between five- and ten-fold within 12 h, with these high levels maintained for at least 24 h. The rising levels of ethylene correlated with rises in expression of the genes encoding *ACS4* and *ACS7* (Figure 4), in the activities of the biosynthetic enzymes *ACS* and *ACO* (Figure 3C,D), and in the accumulation of the intermediate ACC (Figure 3B).

The rise in ABA concentration in treated G. aparine (Figure 5) was as rapid and as extreme as for ethylene. Within 6 h ABA levels had risen by at least three-fold and by five-to ten-fold after 24 h. These rises in ABA correlated well with elevations of expression of the gene coding for the rate-limiting enzyme in biosynthesis, NCED (Figure 6). Taken together, all the data support the hypothesis that the mode of action of halauxifen-methyl in G. aparine is as an SAH, with herbicidal doses leading to rapid and extreme elevations in the expression of genes which code for enzymes that biosynthesize ethylene and ABA [36]. The exogenous SAH also led to rises in endogenous IAA accumulation and this may have contributed to the set of acute responses downstream from auxin perception.

The treatment of G. aparine with halauxifen-methyl has shown that this new herbicide follows the paradigm of SAH activity on dicot weeds. However, it is worth noting that the induction of ACC synthase or ACC oxidase expression has not been observed in all cases and might be dose-dependent. In Arabidopsis, Raghavan et al. (2006) found that the expression of *ACS* and *ACO* was up-regulated after treatment with 0.001 and 0.01 mM 2,4-D, but was not changed after treatment with 0.1 and 1mM 2,4-D [32]. The expression of the *CTR1* gene, a negative regulator of ethylene signaling was down-regulated correspondingly [32,44], which suggested that there was more than one pathway for ethylene to respond to auxin herbicides. McCauley et al. [24] studied the rapid responses to SAHs in Erigeron canadensis using transcriptomics and targeted physiological studies, and found that though ABA accumulation was observed, there was no significant difference in the expression of ethylene biosynthesis genes. Therefore, ethylene synthesis may not be necessary to trigger ABA accumulation in some plants.

In some plants, notably *palaver rhoeas* resistance to applied SAHs has been conferred by reduced ethylene biosynthesis or reduced sensitivity to ethylene which emphasizes the importance of ethylene in the herbicidal responses of some dicot species [45]. Other sources of resistance to SAHs have also been reported including both target site and non-target site resistances [40,46].

In the present study, application of halauxifen-methyl resulted in an increase in IAA content, leading to an imbalance in auxin homeostasis in *Galium aparine*. Consequently, the overexpression of ACC synthase and *NCED* genes in *G. aparine* was induced, and the activity of key rate-limiting enzymes was then enhanced in the ethylene and ABA biosynthesis pathways, respectively, which lead to the overproduction of ethylene and ABA and eventually caused senescence and plant death. Given the specificity of halauxifen-methyl for AFB5 in the auxin receptor family, care must be taken to monitor for weed resistance to this useful new SAH and further studies on its mechanism of action of halauxifen-methyl are needed.

**Author Contributions:** Conceptualization, J.L.; Methodology, J.L. and J.X.; Software, J.X. and X.L.; Validation, J.X. and X.L.; Formal analysis, R.N. and X.L.; Investigation, J.X. and J.L.; Resources, J.L. and L.D.; Data curation, J.X. and X.L.; Writing—original draft, J.X.; Writing—review & editing, J.L., X.L. and R.N.; Visualization, J.X. and X.L.; Supervision, J.L. and L.D.; Project administration, J.L.; Funding acquisition, L.D. and J.L. All authors have read and agreed to the published version of the manuscript.

**Funding:** This research was funded by [National natural science foundation of China] grant number [31871993].

**Institutional Review Board Statement:** Not applicable.

**Informed Consent Statement:** Not applicable.

**Data Availability Statement:** All data from this study are contained within the article.

**Conflicts of Interest:** The authors declare no conflict of interest.

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
