# Peer review of "Mode of Action of a Novel Synthetic Auxin Herbicide Halauxifen-Methyl"

_agronomy, doi:10.3390/agronomy12071659_

Round 1

Reviewer 1 Report

My comments are below:

Remove comma from the title

89 Zobiole did not study auxin MoA. Cite others who did.

Fig 1 changes it to curves instead of bar graphs

Fig 2, 3, 4, and 5 make it colored 

sp name should be italicized 

Missing conclusion

Author Response

Comments and Suggestions for Authors:

Remove comma from the title

Re: Thanks for your suggestion. We have revised the title.

“Mode of action of a novel synthetic auxin herbicide halauxifen-methyl”

89 Zobiole did not study auxin MoA. Cite others who did.

Re: We have cited the research which study on auxin MoA.

“McCauley, C.L., W.G. Johnson, and B.G. Young, Efficacy of Halauxifen-Methyl on Glyphosate Resistant Horseweed (Erigeron canadensis). Weed Science, 2018. 66(6): p. 758-763.”

Fig 1 changes it to curves instead of bar graphs

Re: Thanks for your suggestion and we have changed the figure to curves as your advice.

Fig 2, 3, 4, and 5 make it colored 

Re: Thanks for your suggestion, we have modified these figures.

sp name should be italicized 

Re: We have italicized all sp names.

Missing conclusion

Re: Thanks for your suggestion. The manuscript we submitted include Introduction, Method and Materials, Results and Discussion, Conclusion. It totally four parts in our manuscript.

  We consulted one of editor of Agronomy-Basel, it’s unfortunately that the editor compiled our conclusion into Discussion due to format editing problems. And we organized our manuscript into four parts in this reversion according to the editor’s suggestions, the Conclusion was incorporated into the Discussion, and we have re-written the Discussion. Therefore there would be four parts in the manuscript, including Introduction, Method and Materials, Results, Discussion.

Moderate English changes required

Re: Thanks for your suggestion. The language was enhanced by professional editors at Editage, and we have submitted the editing certificate as supplementary material.

Reviewer 2 Report

In general, the studies are very clear and the results seem consistent.

Author Response

Comments and Suggestions for Authors

In general, the studies are very clear and the results seem consistent.

Re: We are appreciated for your review.

Reviewer 3 Report

The paper was written under the direction of the Agronomy Journal, and fulfills all the criteria for publication. Problems covered by the paper are interesting, with high contribution to the field. Quality of writing it should be commended. Introduction to the problem that was treated in this paper is written clearly and concisely. Well written discussion. The conclusion is missing. Literature data are in line with the topic of the paper.

This manuscript will be acceptable after minor revision (corrections to minor methodological errors and text editing). I have given concrete examples in my comments below.

Specific Comments

Material and methods

1.      Line 151 – delete into

2.      Line 178-180 - All Latin names of weeds should be written italic (refers to the whole text)

Conclusions- line 360

1.      Line 360: Missing?

Patents- line 361-370

1.      Line  363 - Supplementary Materials: missing

2.      Line 364- Author Contributions: missing

Author Response

Comments and Suggestions for Authors

The paper was written under the direction of the Agronomy Journal, and fulfills all the criteria for publication. Problems covered by the paper are interesting, with high contribution to the field. Quality of writing it should be commended. Introduction to the problem that was treated in this paper is written clearly and concisely. Well written discussion. The conclusion is missing. Literature data are in line with the topic of the paper.

This manuscript will be acceptable after minor revision (corrections to minor methodological errors and text editing). I have given concrete examples in my comments below.

Re: We are appreciated for your review. We have modified the manuscript as you suggested.

Specific Comments

Material and methods

1.Line 151 – delete into

Re: We have modified this sentence.

2.Line 178-180 - All Latin names of weeds should be written italic (refers to the whole text)

Re: We have italicized these Latin names.

Conclusions- line 360

Line 360: Missing?

Re: Thanks for your suggestion. The manuscript we submitted include Introduction, Method and Materials, Results and Discussion, Conclusion. It totally four parts in our manuscript.

  We consulted one of editor of Agronomy-Basel, it’s unfortunately that the editor compiled our conclusion into Discussion due to format editing problems. And we organized our manuscript into four parts in this reversion according to the editor’s suggestions, the Conclusion was incorporated into the Discussion, and we have re-written the Discussion. Therefore there would be four parts in the manuscript, including Introduction, Method and Materials, Results, Discussion.

Patents- line 361-370

Line 363 - Supplementary Materials: missing

Re: We have submitted the editing certificate as supplementary material.

Line 364- Author Contributions: missing

Re: We have submitted the web page of author contributions.

Reviewer 4 Report

This is a typical study on the mechanism of a specific herbicide. It has to be noted that the mechanism is not totally unknown as stated by the authors, while the study should be enriched with findings on other weeds as well. Moreover, there is absence of conclusions' section and discussion section is obviously inadequate (poor with very few references). Authors should also put all the latin plant names in italics, report the references numbered inside the text (as required by the journal's format) and also in the references' list have them in the proper an homogenous way. A major revision is strongly recommended.

Author Response

Comments and Suggestions for Authors

This is a typical study on the mechanism of a specific herbicide.

It has to be noted that the mechanism is not totally unknown as stated by the authors, while the study should be enriched with findings on other weeds as well.

Re: We appreciate your suggestion, we found that there are some recent researches study on the MoA of halauxifen‐methyl and we have cited some to enrich our introduction.

“In Brassica napus, halauxifen‐methyl treatment leads to the upregulation of auxin and hormone responses, such as IAA, ABA and ACC concentration [22]. McCauley et al. [23] found that halauxifen‐methyl enhance the expression of NCED and lead to a rapid biosynthesis of ABA in Erigeron canadensis.”

  1. Ludwig-Muller, J., et al., Two Auxinic Herbicides Affect Brassica napus Plant Hormone Levels and Induce Molecular Changes in Transcription. Biomolecules, 2021. 11(8).
  2. McCauley, C.L., et al., Transcriptomics in Erigeron canadensis reveals rapid photosynthetic and hormonal responses to auxin herbicide application. Journal of Experimental Botany, 2020. 71(12): p. 3701-3709.

Moreover, there is absence of conclusions' section and discussion section is obviously inadequate (poor with very few references).

Re: Thanks for your suggestion. The manuscript we submitted include Introduction, Method and Materials, Results and Discussion, Conclusion. It totally four parts in our manuscript.

  We consulted one of editor of Agronomy-Basel, it’s unfortunately that the editor compiled our conclusion into Discussion due to format editing problems. And we organized our manuscript into four parts in this reversion according to the editor’s suggestions, the Conclusion was incorporated into the Discussion, and we have re-written the Discussion. Therefore there would be four parts in the manuscript, including Introduction, Method and Materials, Results, Discussion.

Authors should also put all the latin plant names in italics

Re: Thanks for your suggestion, we have italicized all the latin plant names.

report the references numbered inside the text (as required by the journal's format) and also in the references' list have them in the proper an homogenous way. A major revision is strongly recommended.

Re: Thanks for your suggestion, we have listed the references in the adequate way as the requirement.

Moderate English changes required

Re: Thanks for your suggestion. The language was enhanced by professional editors at Editage, and we have submitted the editing certificate as supplementary material.

Reviewer 5 Report

The Discussion should be more extended and we have seen no conclusions.

line 301, there are no results of ABA production in the no herbicide shoots tissue.

Author Response

Comments and Suggestions for Authors

The Discussion should be more extended and we have seen no conclusions.

Re: Thanks for your suggestion. The manuscript we submitted include Introduction, Method and Materials, Results and Discussion, Conclusion. It totally four parts in our manuscript.

  We consulted one of editor of Agronomy-Basel, it’s unfortunately that the editor compiled our conclusion into Discussion due to format editing problems. And we organized our manuscript into four parts in this reversion according to the editor’s suggestions, the Conclusion was incorporated into the Discussion, and we have re-written the Discussion. Therefore there would be four parts in the manuscript, including Introduction, Method and Materials, Results, Discussion.

line 301, there are no results of ABA production in the no herbicide shoots tissue.

Re: Thanks for your suggestion, the purpose of ABA detect is to show the trend of change, and different treatment with halauxifen‐methyl showed different influences after treated by halauxifen‐methyl.
